# A Water Quality Prediction Method Based on the Deep LSTM Network Considering Correlation in Smart Mariculture

**DOI:** 10.3390/s19061420

**Published:** 2019-03-22

**Authors:** Zhuhua Hu, Yiran Zhang, Yaochi Zhao, Mingshan Xie, Jiezhuo Zhong, Zhigang Tu, Juntao Liu

**Affiliations:** 1State Key Laboratory of Marine Resource Utilization in South China Sea, College of Information Science & Technology, Hainan University, No.58, Renmin Avenue, Haikou 570228, China; eagler_hu@hainu.edu.cn (Z.H.); hnmingshanxie@163.com (M.X.); zjz@hainu.edu.cn (J.Z.); 20151681310143@hainu.edu.cn (J.L.); 2School of Software & Microelectronics, Peking University, No.24, Jinyuan Road, Daxing District, Beijing 102600, China; 1801210740@pku.edu.cn; 3Hainan Academy of Ocean and Fisheries Sciences, No.12, Baiju Avenue, Haikou 571126, China; swnurabbit@126.com

**Keywords:** aquaculture water quality, smart mariculture, LSTM deep learning, Pearson’s correlation coefficient

## Abstract

An accurate prediction of cage-cultured water quality is a hot topic in smart mariculture. Since the mariculturing environment is always open to its surroundings, the changes in water quality parameters are normally nonlinear, dynamic, changeable, and complex. However, traditional forecasting methods have lots of problems, such as low accuracy, poor generalization, and high time complexity. In order to solve these shortcomings, a novel water quality prediction method based on the deep LSTM (long short-term memory) learning network is proposed to predict pH and water temperature. Firstly, linear interpolation, smoothing, and moving average filtering techniques are used to repair, correct, and de-noise water quality data, respectively. Secondly, Pearson’s correlation coefficient is used to obtain the correlation priors between pH, water temperature, and other water quality parameters. Finally, a water quality prediction model based on LSTM is constructed using the preprocessed data and its correlation information. Experimental results show that, in the short-term prediction, the prediction accuracy of pH and water temperature can reach 98.56% and 98.97%, and the time cost of the predictions is 0.273 s and 0.257 s, respectively. In the long-term prediction, the prediction accuracy of pH and water temperature can reach 95.76% and 96.88%, respectively.

## 1. Introduction

In smart mariculture, it is an inevitable trend for aquaculture to become smarter, more accurate, and more ecological. However, due to the influence of climate, typhoons, rain, and changes of culture density of seawater, the balance of algae and bacteria in an aquaculture environment can easily be destroyed. Consequently, this leads to a decrease in the anti-stress ability and disease-resistant ability of farmed fish [1,2,3,4]. Furthermore, in traditional mariculture, water quality can only be determined by breeding workers using their experiences, although it is often impossible to grasp the changing trend of water quality in a timely and accurate manner based on empirical judgment alone. The precise prediction of water quality parameters can help aquaculture farmers to get hold of the trend of water quality parameters in the future, so as to adopt countermeasures. Therefore, it is necessary to figure out an accurate prediction method for dynamic changes in water quality factors, which considers the system dynamics relationships between water quality parameters. 

The dynamic changes of water quality parameters involved in different stages of mariculture are extremely complicated. The parameters frequently considered in mariculture are salinity, water temperature, pH, dissolved oxygen, hydro chemical factors, etc. [5]. In this paper, pH and water temperature prediction are studied.

Seawater is the main medium for the habitat and material exchange of aquaculture organisms in mariculture, while aquaculture organisms are sensitive to changes in physical and chemical factors of water. Firstly, large changes in dissolved oxygen, temperature, pH, and other water quality factors may directly lead to the death of aquaculture organisms. Even small fluctuations outside the optimal conditions may cause physiological stress on organisms, such as reduced food intake, increased energy consumption, and susceptibility to infectious diseases. Additionally, in mariculture the aquaculture environment is also an artificial ecosystem. Changes in water quality parameters also affect zooplankton, phytoplankton, and various bacteria in the water environment, which may lead to the deterioration of the aquaculture ecological environment, such as an outbreak of red tide algae, bacteria, and parasites. 

The above-mentioned changes in water quality will result in a decrease in aquaculture production efficiency. Therefore, if the changing trend in water quality can be predicted, we can take countermeasures in advance through technical means to prevent serious imbalances in the water quality environment. It can be seen that an accurate prediction of water quality can greatly improve the production efficiency of mariculture.

Since water quality data is often preprocessed before the water quality parameters are predicted, this section reviews from two stages.

The first stage is the pretreatment of water quality data and correlation analysis between different water quality parameters. In the field of data restoration, scholars have done a lot of research. Jin et al. [6] designed a new data repairing algorithm based on functional dependency and conditional constraints, which improved the efficiency of data restoration to a certain extent. Zhang et al. [7] constructed an observation matrix based on compressed sensing and a dictionary matrix combined with the prior knowledge to sparsely represent data. The data is then accurately repaired from lossy data, the observation matrix, and data repaired by the dictionary matrix. Xu et al. [8] proposed a fault data identification method based on a smoothing estimation threshold method and a fault data repair method based on statistical correlation analysis, which improved the repairing accuracy. Singh et al. [9] proposed a data recovery algorithm based on an orthogonal matching pursuit algorithm for infinite sensor networks, which significantly reduced the number of iterations needed to repair the original data within a small interval. Jakóbczak et al. [10] presented a probabilistic feature combination method, which uses a set of high-dimensional feature vectors for multi-dimensional data modeling, extrapolation, and interpolation; this method combines the numerical method with the probabilistic method and achieves high accuracy. A great deal of effort has also been done by researchers in the field of data noise reduction. Zhang et al. [11] put forward a data de-noising method based on an improved pulse coupled neural network, which is more suitable for high-dimensional data denoising. Pan et al. [12] used wavelet technology to denoise, which not only achieved denoising, but also retained the characteristics of the data itself. Wang et al. [13], Li et al. [14], and Perelli et al. [15] have also used the same method. In our research, due to the short sampling interval and small error data amount, the linear interpolation method and smoothing method can be used to complete interpolation and error correction. In addition, since the noise in water quality data is mainly caused by water fluctuation, the moving average method is used to filter noise. 

For correlation analysis, the correlation of two random variables can be well measured by Pearson’s correlation coefficient method, which is divided by the standard deviation of two random variables on the basis of covariance. However, when the amount of data to be analyzed is insufficient, the analysis results are unreliable. Recently, some new methods of correlation analysis have been proposed, such as the spatial cross-correlations method [16]. This method combines spatial distribution information and uses semivariance and experimental variograms to calculate the spatial correlation of water quality. Then, cross-correlations are analyzed using experimental cross-variograms. The water quality in a small lake can be described well using this method.

The second stage is water quality prediction using the pre-processed data. Traditional water quality prediction methods mainly include: the Grey Markov chain model method [17,18], the fuzzy-set theory-based Markov model [19], the regression prediction method [20,21,22,23], the time series method [24,25], and the water quality model prediction method [26,27]. The water quality model prediction method has poor self-adaptability, and the other traditional prediction methods also have many shortcomings, such as low prediction accuracy, poor stability, and single factor prediction without considering dynamics characteristics. With the development of computational intelligence and bionic technology, many novel prediction methods based on artificial intelligence have emerged. The new water quality prediction methods mainly include: the grey theory method [28], the artificial neural network method [29,30,31], the least squares support vector regression prediction method [32,33], the combination prediction method [34], etc. These methods provide an effective solution for water quality prediction, but are still not perfect due to the fact that mariculture environments are affected by many factors. These factors are mainly reflected in the complex interaction mechanism between water environment parameters, the non-linearity of water quality changes, and the long delay, which lead to a low calculation efficiency and poor generalization performance in the mentioned prediction methods. 

In order to solve the problems mentioned above, combined with preprocessing and correlation analysis, the key water quality parameters (pH and water temperature) prediction models based on an LSTM (long short-term memory) [35] deep neural network are trained. For correlation analysis, Pearson’s correlation coefficient method is adopted because we have obtained sufficient data and have only one sensor deployment location. LSTM is improved from RNN (recurrent neural network) [36]. RNN is usually used to process time sequential data [36]. This kind of data reflects the state or the changing degree over time of something, a phenomenon, etc. Then, the training prediction models are used to accurately predict pH and water temperature. At last, the prediction model is evaluated. 

Our main contributions can be summarized as follows:The linear interpolation method and smoothing method are used to fill and correct the data sampled by the sensors, respectively. The moving average filter is used to denoise the data after filling and correcting.The influence factors of pH and water temperature are analyzed comprehensively. The correlation between water temperature, pH, and other water quality parameters is obtained by Pearson’s correlation coefficient method, which can be used as the input parameters of the model training.Based on the pre-processed data and the correlation analysis results, a water quality prediction model based on a deep LSTM learning network is trained. Compared with the RNN based prediction model, the proposed prediction method can obtain higher prediction accuracy with less time.

The rest of this paper is organized as follows. Section 2 gives the methods of data acquisition and data analysis, and presents the water quality prediction model based on LSTM. In Section 3, we analyze and discuss the experimental results of pretreatment, and evaluate the accuracy and time complexity of proposed prediction methods for pH and water temperature. Finally, Section 4 concludes this paper.

## 2. Materials and Methods

### 2.1. Data Acquisition

The experimental data was collected in the mariculture base at Xincun Town, Lingshui County, Hainan Province, China. Data collection was achieved by deploying sensor devices in a cage. The collected data was stored in the data server, and real-time data information could be viewed on the mobile terminal. Figure 1a shows the culture cage. It can be seen that a small amount of dead fish has floated above the water because of the deterioration in the water quality. Figure 1b shows a data acquisition sensor device, a vertical axis wind power generation device, a solar power generation panel, a storage battery, and a wireless transmitting device. Figure 1c shows the online data monitoring display page on a mobile phone. 

### 2.2. Data Preprocessing

High-quality sample data sets are the basis of accurate analysis and prediction. In wireless sensor networks, however, due to instability, aging or erosion of the sensor equipment, and the susceptibility of the transmission network to distance and the surrounding environment, data loss, abnormality, and noise interference may occur during the measurement and transmission of water quality parameters [37].

#### 2.2.1. Data Filling and Correction

In some application scenarios in this research where water quality changed dramatically, the linear interpolation was used, since it has better robustness than nearest-neighbor interpolation and is also suitable for filling data sets with small data intervals. The relationship between two known data and one unknown datum is seen as a linear relationship by the linear interpolation method, which uses the slope of the assumed line to calculate the data increment, thereby obtaining the required unknown data. This method is shown in (1): (1)$$x_{k+i}=x_{k}+\frac{i\cdot (x_{k+j}-x_{k})}{j},$$
where $x_{k+i}$ denotes lost data, $x_{k}$ is theknown data before $x_{k+i}$, $x_{k+j}$ is the jth known data after $x_{k}$.

Water quality data are usually continuous, and normally change slowly and smoothly, with no sudden increase or decrease in a very short period of time. Therefore, if the received water quality data has a sharp jump, it needs to be corrected. This paper applied the smoothing method on data error correction. Error correction involves two steps: error detection and correction. If the error is detected, it is necessary to check whether the relative difference between the current data and the previous data or the latter data exceeds a certain range. If it does, the current data is wrong, otherwise the data is correct. The data correction process is to use the mean value of the previous and latter data of the erroneous data as the correction value, and the erroneous data is then replaced with the correction value. This method is shown in (2):(2)$$x_{k}=\left\{ \begin{array}{l} \frac{x_{k-1}+x_{k+1}}{2}\;\mathrm{if}\;\left|x_{k}-x_{k-1}\right|>\beta_{1}\;\mathrm{or}\;\left|x_{k}-x_{k+1}\right|>\beta_{2}\\ x_{k}\;\mathrm{else} \end{array}\right.,$$
where $\beta_{1}$ and $\beta_{2}$ are adjacent data error thresholds, and $x_{k}$ represents the data currently needed for error detection. If the difference between $x_{k}$ and $x_{k-1}$ is larger than $\beta_{1}$ or the difference between $x_{k}$ and $x_{k+1}$ is larger than $\beta_{2}$, then it has to be updated as the mean of the two values before and after.

#### 2.2.2. Moving Average Filtering

In the complex aquaculture environment there are noise interference signals in water quality data, for example, noise caused by constantly oscillating water waves. Figure 2 shows that the noise frequency in the water quality data was relatively stable. Therefore, the moving average filter can be used to achieve data noise reduction. 

The moving average filter is a finite-length unit impulse response filter, shown as (3).
(3)$$\begin{array}{ll} y(t) & =\frac{1}{N}[x(k)+x(k-1)+\ldots +x(k-N+1)]\\ & =\frac{1}{N}\sum_{k=0}^{N-1}x(t-k) \end{array}$$

In (3), N represents the size of window, t is a certain time, y(t) is the data in the new sequence corresponding to time t (the average value of N data), and x(t−k) is the data at time (t−k) in original data set. The moving average filtering can attenuate high frequency signals, so as to achieve data smoothing.

### 2.3. Correlation Analysis

In this paper, the Pearson’s correlation coefficient method was used to analyze the correlation between water temperature, pH, and other water quality in an aquaculture environment. Pearson’s correlation coefficient [38] is a method for analyzing wheth er there is a close correlation between two variables, which is defined as the quotient of the covariance and the standard deviation between two variables. After pretreatment of all measured water quality parameters, the Pearson’s correlation coefficient method was used to analyze the correlation between the required predicted water quality factors and other factors. The calculation results are shown in Table 1.

Table 1 shows that the water temperature has a strong positive correlation with conductivity, a strong negative correlation with salinity, a moderate positive correlation with chlorophyll, a weak correlation with turbidity, an extremely strong negative correlation with pH, and a strong positive correlation with dissolved oxygen. On the other hand, pH has a strong negative correlation with conductivity, a strong positive correlation with salinity, almost no correlation with turbidity, and a weak correlation with dissolved oxygen.

### 2.4. The Proposed Prediction Model Based on LSTM Deep Learning

Water quality prediction plays a significant role in smart aquaculture. The trends of changes in water quality in the future are predicted using historical water quality data and the correlations with other water quality. If the predicted water quality parameters are beyond the tolerance range of the fish, the water quality should be regulated in time to keep the fish living in the most suitable water environment for a long time.

#### 2.4.1. LSTM Deep Learning Network

The LSTM neural network is an improvement on the recurrent neural network (RNN). The main difference is that a processor called a “cell state” is added to the hidden layer, which is used to determine whether the information is useful or not. The advantage of the RNN is that it can continuously remember historical information during the training along the sequence, however its memory capacity is limited. As the training progresses, artificial neurons forget information that is considered irrelevant and only remember some information that is considered important, which leads to the loss of some information needed at the later stage. However, the LSTM neural network has been improved to resolve this problem.

The hidden layer of the traditional RNN has only one state *s*, while the LSTM network added a state c, that is, cell state. As shown in Figure 3, at time t, the hidden layer has three inputs, namely, the input value $x_{t}$ at the current time, the output value $s_{t-1}$ of the hidden layer neurons at the previous time, and the unit state $c_{t-1}$ at the previous time. Meanwhile, there are two outputs in the hidden layer, namely, the output of the hidden layer $s_{t}$ and the cell state $c_{t}$ at the current time. 

In Figure 4, the LSTM network sets up three gates to control c, namely, $r_{1}$ (called the forget gate, which is used to control whether to save the long-term state), $r_{2}$ (called the input gate, which is used to control whether the current state is inputted into the long-term state), and $r_{3}$ (called the output gate, which controls whether the current long-term state is the output of the hidden layer). The forget gate determines how much information of state $c_{t-1}$ at the previous moment is retained to the current state $c_{t}$. The input gate determines how much information of data $x_{t}$ inputted into the hidden layer in current is saved to the current state $c_{t}$. Similarly, the output gate controls how much information in $c_{t}$ is inputted into $s_{t}$. The internal structure of the hidden layer in the LSTM network is shown in Figure 5. 

In Figure 5, $x_{t}$ is the input data set, $s_{t-1}$ is the output of previous hidden layer, $f_{t}$ is the forget gate, $W_{f}$ is the weight of the forget gate, $i_{t}$ is the input gate, $W_{i}$ is the weight of the input gate, $c_{t}^{\prime }$ is the input cell state at the current time, $W_{c}$ is the weight of $c_{t}^{\prime }$, $o_{t}$ is the output gate, $W_{o}$ is the weight of the output gate, $c_{t-1}$ is the previous unit state, $c_{t}$ is the current unit state, and $s_{t}$ is the output of the current hidden layer. 

The LSTM network has the process of data back-propagation, the same as the RNN, and the error value propagates along the time series in addition to spreading between layers. After obtaining the updated gradient of the horizontal and vertical weights and bias terms, the updated value of each weight and bias term can be obtained by the structure of the hidden layers. The calculation method is the same as the RNN network, and the value of the learning rate *α* should be set to control the error updated gradient and the speed of error decline.

In the above training model, we introduced three evaluation metrics [39] to evaluate the prediction effect, which are defined as follows:

**Definition** **1:**
*MAE (mean absolute error)*

*MAE is the basic evaluation metric, and the following methods are generally used as a reference to compare the advantages and disadvantages.*
(4)$$MAE=\frac{1}{N}\sum_{i=1}^{N}\left|y_{i}-{\bar{y}}_{i}\right|.$$


**Definition** **2:**
*RMSE (root mean squared error)*

*RMSE denotes the mean error, which is more sensitive to extreme values. If there is an extreme value in the training process at some time point, the RMSE will be greatly affected by the increasing error. The change of the evaluation index can be used as the benchmark for the robustness test of the model.*
(5)$$RMSE=\sqrt{\frac{1}{N}\sum_{i=1}^{N}{\left(\left|y_{i}-{\bar{y}}_{i}\right|\right)}^{2}}.$$


**Definition** **3:**
*MAPE (mean absolute percent error)*

*MAPE considers not only the deviation between the predicted data and the real data, but also the ratio between the deviation and the real data.*
(6)$$MAPE=\frac{1}{N}\sum_{i=1}^{N}\frac{\left|y_{i}-{\bar{y}}_{i}\right|}{y_{i}}.$$

*In (4), (5), and (6),*
$y_{i}$
*represents the real value,*
${\bar{y}}_{i}$
*represents the value predicted by the model at the same time, which is the output value of the deep learning model, and N is the number of samples in the data set. The closer the above three evaluation metrics are to 0, the better the prediction and fitting effect of the model will be.*


#### 2.4.2. Construction of a Water Temperature and pH Prediction Model Based on the LSTM Deep Network

The whole prediction process is shown in Figure 6. The proposed prediction model integrates data preprocessing and correlation analysis into the training of the deep LSTM network. Firstly, after receiving water quality data from the wireless transmission network, a series of linear interpolation, smoothing, and moving average filtering techniques are used to repair, correct and de-noise water quality data, respectively. Then, Pearson’s correlation coefficient is used to obtain correlation priors between pH, water temperature, and other water quality parameters. Finally, a water quality prediction model based on deep LSTM is constructed using the preprocessed data and its correlation information. When the prediction accuracy of the model reaches the expected requirements, the whole prediction model is considered to be established successfully; otherwise, it will be retrained to obtain better results.

According to the system dynamics relationship between the water quality factors analyzed in Section 2.3, it is easy to know that water temperature has a strong correlation with conductivity, salinity, pH, and dissolved oxygen. The historical data of these parameters and water temperature were used as the input data of the water temperature prediction model constructed for training, and the output of the model was the predicted value of water temperature. The actual water temperature data measured locally was taken as the real data.

Using the TensorFlow platform and python language, the prediction models based on the RNN and the LSTM deep network were built. The preprocessed data of 610 groups were input into the prediction model separately along a time series. In the water temperature prediction model, the input data dimension was 5, the output data dimension was 1, the number of hidden layers was 15, the time step was set as 20, the learning rate was set to 0.0005, and the times of training were set as 10,000. In each training process, three evaluation metrics—MAE, RMSE, and MAPE—between the output values of all output layers and the real values were recorded, as shown in Table 2 below. 

The comparison of RMSE (water temperature) between the LSTM-based prediction model and the RNN-based prediction model is shown in Figure 7. For water temperature, the unit of the RMSE is Celsius (°C).

In the same way, according to the analysis in Section 2.3, it can be seen that pH has a strong correlation with conductivity, salinity, and water temperature. These water quality parameters and the historical data of pH were used as the input data for the model construction, the output value of the model was taken as the predicted value of the pH, and the actual measured pH data was taken as the real data. When building the prediction model, the dimension of the input data was 4, the dimension of the output data was 1, the hidden layer was set to 15 layers, the time step was set to 20, the learning rate was set to 0.0005, and the training was performed 10,000 times. The three evaluation metrics between the output values of all output layers and the real value in each training process was recorded, as shown in Table 2. The comparison of RSME (pH) between the LSTM-based prediction model and the RNN-based prediction model is shown in Figure 8.

From Figure 7a and Figure 8a, we can see that the error between the output data and the real data was gradually reduced, and finally was close to 0. In addition, in the early stage of training, the error slowed down faster, while the dropping rate gradually slowed down in the middle and late stages. Figure 7b and Figure 8b are the charts of partial magnification comparison of error. It can be seen that the overall trend of the error was decreasing. Since the error of the entire training network was larger each time when the training started again, a local rising phenomenon occurred. However, as the single training went on, the error kept decreasing. 

## 3. Results and Discussions

The experimental data was collected from the mariculture cages with the sensor devices, and then transmitted to the data server by means of a wireless bridge for storage. For the short-term predictions, the sampling frequency of the data was once every 5 min. Water quality data of 610 groups (about 51 h) including temperature, conductivity, chlorophyll, salinity, turbidity, pH, and dissolved oxygen parameters were used as experimental data for model training, and another 100 sets of water quality data (about 8.3 h) were used to verify the prediction effect. In addition, for the long-term predictions, the data sampling interval and data collection quantity are described in Section 3.4.

The experimental environment is: Intel(R) Core(TM) i7-8550 CPU@2Ghz processor, 8 GB memory, Windows 10(64-bit) operating system, Anaconda3 experimental platform, and pycharm3.3 IDE (Integrated Development Environment), and the construction of the neural network model is based on python 3.6 and the Tensorflow 1.6.0 package. The accuracy and range ability of the sensors are shown in Table 3. F.S. is the abbreviation for “Full Scale”, NTU is the abbreviation for “Nephelometric Turbidity Unit”, and PSU is the abbreviation for “Practical Salinity Unit”.

### 3.1. Experiments and Analysis of Data Preprocessing

Compared with spline interpolation, nearest-neighbor interpolation, and cubic interpolation, it has been found that linear interpolation has a similar interpolation effect to nearest-neighbor interpolation, and is superior to spline interpolation and cubic interpolation. Therefore, in this experiment, we used the improved method mentioned in Section 2.2.1 for data filling.

In the process of data mending, taking water temperature data collected at depth of 3.26 m as an example, in order to determine the optimal value of i and j in (1), the relative error between the original data and filling data obtained by the linear interpolation method is calculated when i and j are both positive integers in interval [1, 10]. The variation is shown in Figure 9.

As shown in Figure 9, in the deep orange area—i.e., i∈[1,4]∩j∈[1,10], i∈[4,6]∩j∈[4,10], or i∈[6,10]∩j∈[6,10]—the relative errors between the original data and filling data are nearly 0, while, in the red area, blue area, and the area near them—i.e., i∈[4,10]∩j∈[1,4]—the relative errors are close to 0.04. Furthermore, as can be seen from Figure 9, the relative errors can be minimized when (i∈[1,4]∩j∈[7,10])∪(i∈[6,10]∩j∈[7,10]).

In terms of data correction, since water quality data have a time correlation, $\beta_{1}$ and $\beta_{2}$ in Equation (2) can be determined using the relative difference between two adjacent historical water quality data as a constraint of the current relative difference. Take the average value of the relative difference between two adjacent data of the previous day as the value of $\beta_{1}$ and $\beta_{2}$, i.e., $\beta_{1}=\beta_{2}=10\%$, the relative differences of pH and water temperature before and after data correction are shown in Figure 10.

As shown in Figure 10, red and blue dots overlap together, and the relative differences of temperature and pH before and after data correction are not greatly different, which indicates that there are relatively few error data in the collected data.

In the process of data denoising, Equation (3) is used in the experiment. The size of window N was set as 4, the data of water temperature and pH were smoothed and denoised. Comparisons of water quality data before and after denoising are shown in Figure 11.

From Figure 11, it can be seen that the moving average filter can effectively reduce the data noise, restore the original data affected by wave and transmission, and smooth the water quality parameter curve.

### 3.2. Experiments and Analysis of Water Temperature Short-term Prediction

#### 3.2.1. The Prediction of Water Temperature

Two kinds of prediction models were used to predict the variation trend of water quality parameters in the future. The 100 values were predicted, and the comparison between predicted values and real values is shown in Figure 12. 

The water temperature data predicted by the two models mentioned above is not completely matched with the real value, but the value predicted by the LSTM-based model is closer to the real value. Obviously, the values predicted by the RNN-based prediction model fluctuate greatly, and the errors between the predicted value and the real value are also large. Table 4 shows the relative deviations between the predicted values and the real values of the two models. The unit of the deviation in LSTM or RNN is degrees Celsius. In order to facilitate typesetting, the 100 groups of deviation values between the predicted data and the real data are divided into four columns from left to right, with each column showing 25 data.

In Table 4, relative deviations between the predicted values and the real values using the model based on LSTM are mostly less than 1 °C, with an average of 1.03 °C, while the deviations using the model based on RNN are mostly more than 1 °C, with an average of 1.37 °C. As a result, the LSTM-based model can predict water temperature more effectively and more accurately.

#### 3.2.2. Time Complexity Analysis

The duration of each training and the total time cost of 10,000 times training under the two neuron networks were recorded in the experiment. Figure 13 shows a comparison of the time spent performing 10,000 trainings between the two methods. 

From Figure 13, it can be seen that the training time of the LSTM-based prediction model is shorter and more stable, while the training time of the RNN-based model is longer, and increases sharply between the 7000th and 9000th. The average training time of the LSTM neuron network is 0.257 s, and the total time cost of 10,000 times training is 2567.06 s; the average training time of RNN is 0.259 s, and its total training time is 2591.95 s. Therefore, the training time of the LSTM-based prediction model is shorter than that of the RNN-based model. In other words, the construction efficiency of the LSTM-based prediction model is higher.

### 3.3. Experiments and Analysis of pH Short-Term Prediction

#### 3.3.1. Prediction of pH Values

The future pH data is predicted using the two trained models. The 100 values are predicted, and the comparison between the predicted values and real values is shown in Figure 14.

Figure 14 shows the predicted contrast effect after the scale of the vertical axis is enlarged. In fact, the relative errors are no more than 5%, and the future trend of pH can be judged from Figure 14. The predictions based on the LSTM deep network are closer to the real values. Table 5 shows the relative deviations between the predicted values and the real values under the two models. In order to facilitate typesetting, the 100 groups of deviation values between the predicted data and the real data are divided into four columns from left to right, with each column showing 25 data.

As shown in Table 5, the average relative deviation between the predicted values and the real values using the RNN-based prediction model is 1.579, while the one using the LSTM-based model is 1.439. Therefore, the predicted results of the LSTM-based model are closer to the real values.

#### 3.3.2. Time Complexity Analysis

The experiments recorded the time cost of each training and the total time consumption of 10,000 training times for pH data using two prediction models. Figure 15 shows the comparison of 10,000 training times between the two methods. 

As can be seen from Figure 15, the training time of the LSTM network is apparently shorter than that of RNN, and the time variations of the former are also smaller, thus it is more stable when using the LSTM network. The average training time of the RNN is 0.298 s, and the total time cost of 10,000 times training is 2968.568 s, while the average training time of the LSTM network is 0.273 s, and its total training time is 2734.118 s. Therefore, the LSTM network takes less time and is more efficient in constructing the pH prediction model.

### 3.4. Long-Term Prediction of Water Temperature and pH

In order to further verify the practicability and robustness of the prediction model, a longer training data set was collected for model training. Then, we used the trained model to predict the next 83 h (about 3.5 days) of water quality data. The sampling frequency of the data is once every 1 min. A total of 30000 groups (about 21 days) of data were collected for training. An additional 5000 sets of data (83 h in total) were used for comparison. 

The experiment was carried out under the same conditions as the short-term prediction, and the number of trainings was 500 and 1000, respectively. The results of the three evaluation indicators obtained during each training process are shown in Table 6.

For the water temperature prediction, the comparison of RMSE between the LSTM-based prediction model and the RNN-based prediction model is shown in Figure 16. According to equation (5), when RMSE is closer to 0, the prediction error of water quality parameters is smaller. For water temperature, the unit of the RMSE is Celsius (°C).

For the pH prediction, the comparison of RMSE between the LSTM-based prediction model and the RNN-based prediction model is shown in Figure 17. 

The trained model was used to predict the water temperature and pH values. We have predicted a total of 5000 sets of data. The comparison of the long-term prediction effect between the proposed scheme and RNN is shown in Figure 18 and Figure 19. It takes a total of 66 s to predict 5000 pieces of data using the trained model, and the average prediction time is 13.2 ms.

Different kinds of fish have different tolerances to water quality parameters. Saddle-spotted grouper (*epinephelus lanceolatus*) for example, generally has a pH value tolerance range of 7.5–9.2, and the range suitable for growth is 7.9–8.4. The water temperature suitable for growth is 22 °C–30 °C, with a minimum tolerance of 15 °C and a maximum tolerance of 35 °C. Because cultured fish are sensitive to changes in key water quality parameters, countermeasures can be taken in advance through water quality prediction to keep water quality parameters within the tolerance threshold range.

From Figure 18 and Figure 19, we can see some spikes. Since these spikes don’t last very long, we treat them as predicted abnormal data without intervention. However, if such spikes last longer (i.e., more than 15 min) and are outside the tolerance threshold, the farmer needs to pay close attention and take countermeasures in advance. 

### 3.5. Discussions

From the above experimental analysis, the proposed scheme can achieve better results in long-term and short-term prediction. Using the proposed scheme, the short-term prediction accuracy can reach 98.56% and 98.97% for pH and water temperature, respectively, while the long-term prediction accuracy can reach 95.76% and 96.88% for pH and water temperature, respectively. In addition, the average prediction time for short-term predictions is 12.5 ms, and the average time for long-term predictions is 13.2 ms. Therefore, based on the trained model, the proposed scheme can realize fast and accurate predictions.

However, the proposed scheme still needs more computational cost in data set processing. Moreover, compared with the real data, the overall prediction results have strong fluctuations. In the future, we will focus on the optimization of the deep learning network structure. On the premise of ensuring the prediction accuracy, we will reduce the computational complexity of model training through optimization. Meanwhile, in order to make the water quality prediction model more robust and practical, the deep neural network structure will incorporate more relevant prior knowledge (such as precipitation and climate factors) for prediction. In addition, the proposed method also has some limitations in data preprocessing. According to the SVD (Singular Value Decomposition) theory [40,41], we know that the original signal contributes little to the tail singular values, and the signal energy is mainly concentrated on the first several singular values, while the tail singular values is mainly determined by noise. In future work, we will obtain more effective noise reduction methods based on this conclusion. Meanwhile, for some of the meaningless spikes that appear in Figure 18 and Figure 19, we will consider how to conduct reasonable and safe post-processing in our future work to make the prediction curve smoother.

## 4. Conclusions

Aiming at the problems of water quality prediction in smart mariculture, a relatively accurate water quality prediction method based on the deep LSTM network is proposed, which integrates the correlation coefficient between water quality data. In the proposed method, the integrity and accuracy of data are effectively improved after data pretreatment operation in the first stage. Then, the correlation knowledge between pH, water temperature, and other water quality parameters is obtained using Pearson’s correlation coefficient method. Ultimately, the water quality prediction model based on the deep LSTM network is constructed. The prediction results of pH and water temperature using the constructed prediction model show that the proposed method can achieve a higher prediction accuracy and lower time cost than the RNN-based prediction model. Specifically, for the short-term predictions, the prediction accuracy of the proposed scheme can reach 98.56% and 98.97% in the prediction of pH and water temperature, respectively, while, for long-term predictions, the prediction accuracy of the proposed scheme can reach 95.76% and 96.88%.

## Figures and Tables

**Figure 1 sensors-19-01420-f001:**
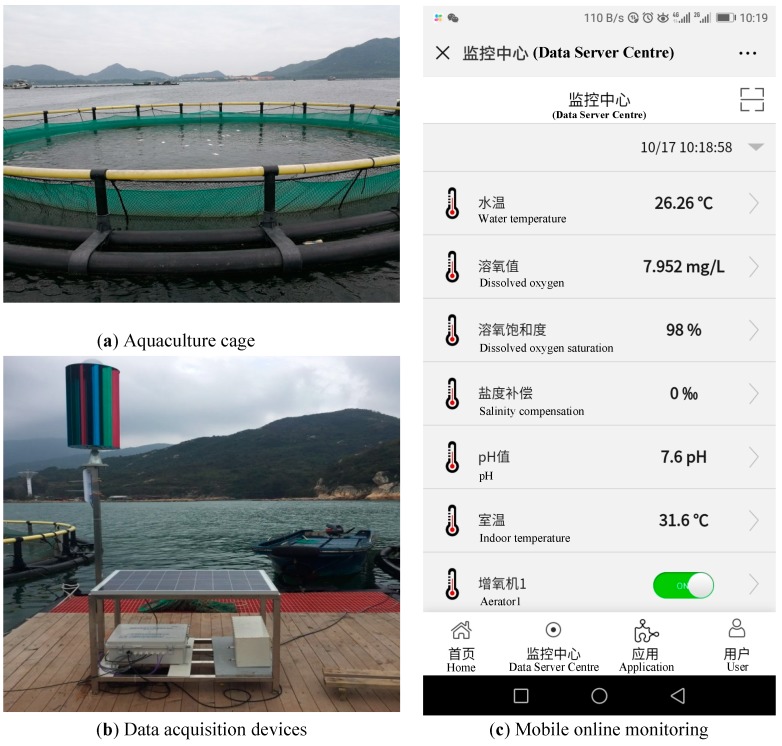
Experimental data acquisition site and software.

**Figure 2 sensors-19-01420-f002:**
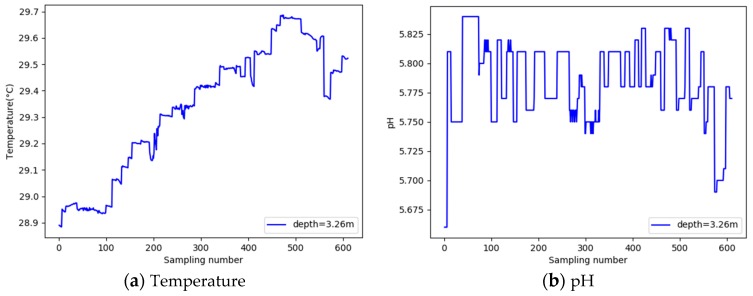
Water quality parameters vary with the number of samples.

**Figure 3 sensors-19-01420-f003:**
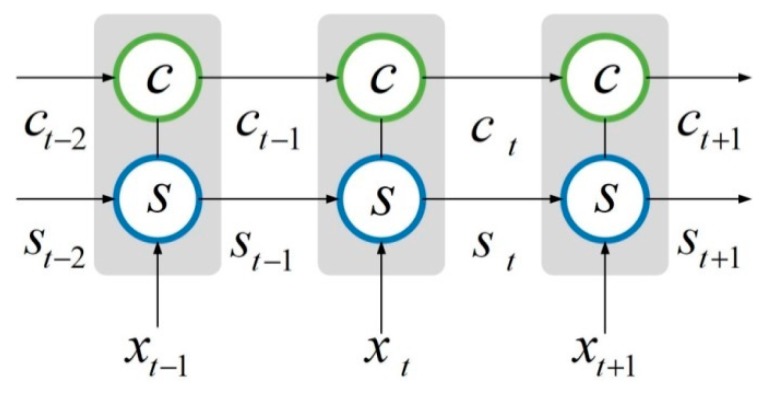
Schematic diagram of the hidden layer of the LSTM neural network.

**Figure 4 sensors-19-01420-f004:**
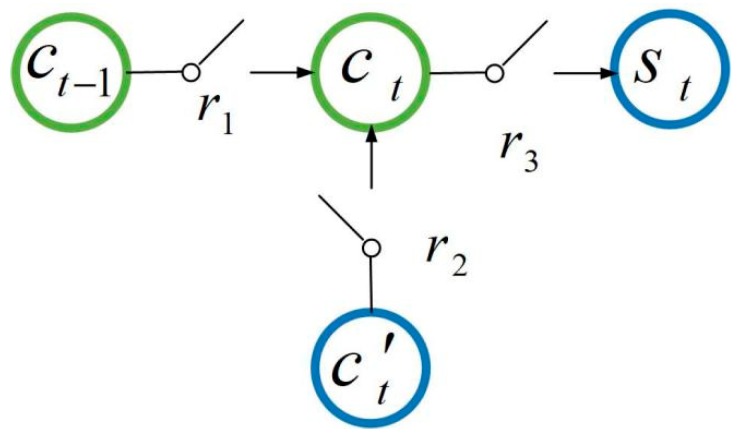
Three gates of the LSTM network.

**Figure 5 sensors-19-01420-f005:**
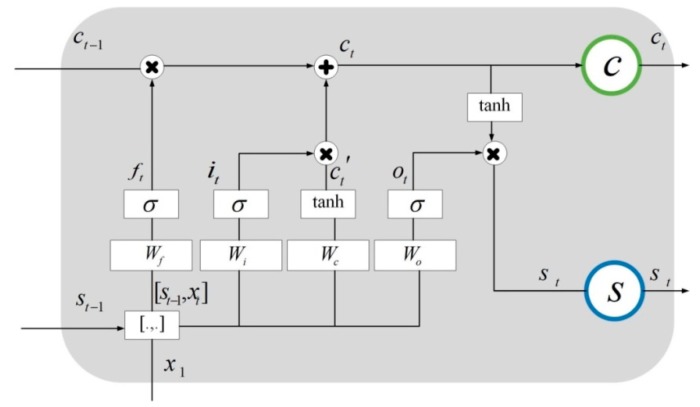
The internal structure of the hidden layers of the LSTM network.

**Figure 6 sensors-19-01420-f006:**
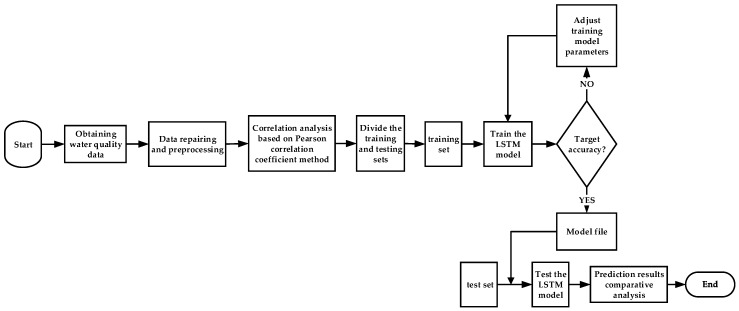
The complete flowchart of the water quality prediction model.

**Figure 7 sensors-19-01420-f007:**
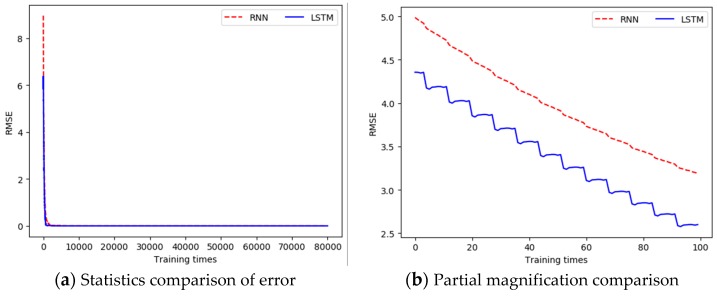
Comparison of the difference between the output value in each training and the real value when building the water temperature prediction model.

**Figure 8 sensors-19-01420-f008:**
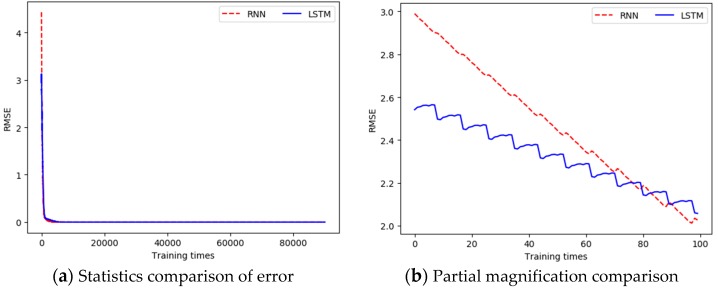
Comparison of the error between the pH output value and the real value in each training.

**Figure 9 sensors-19-01420-f009:**
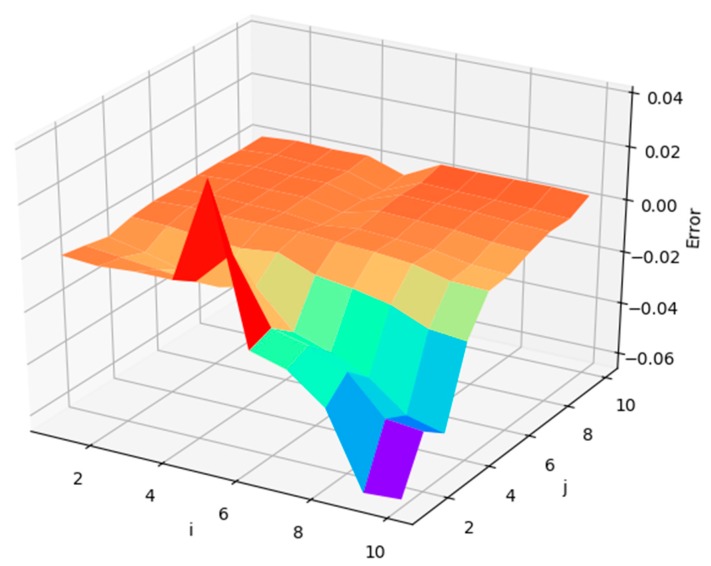
Changes of relative error between original data and filling data with the variations of i and j.

**Figure 10 sensors-19-01420-f010:**
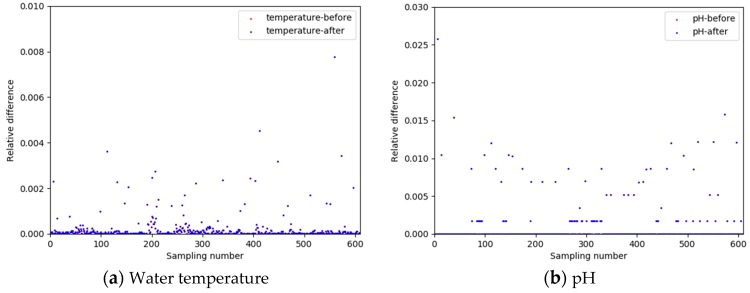
Comparison of relative differences before and after data correction.

**Figure 11 sensors-19-01420-f011:**
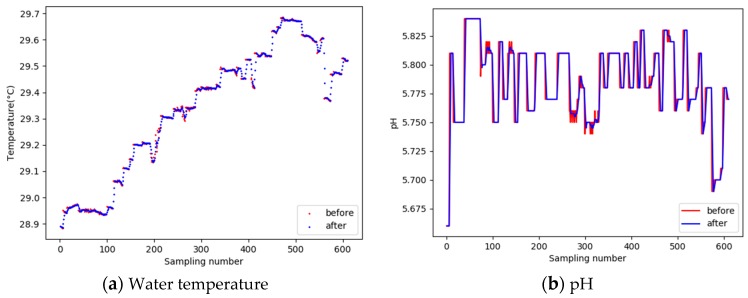
Comparison of water quality data before and after noise reduction.

**Figure 12 sensors-19-01420-f012:**
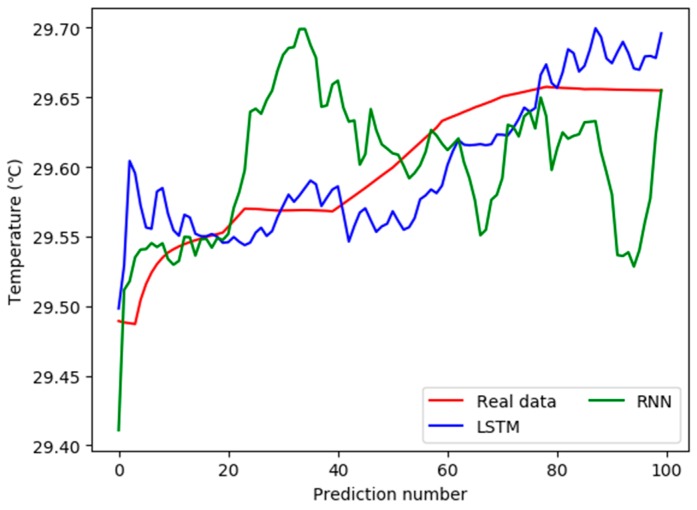
Comparison between predicted values and real values.

**Figure 13 sensors-19-01420-f013:**
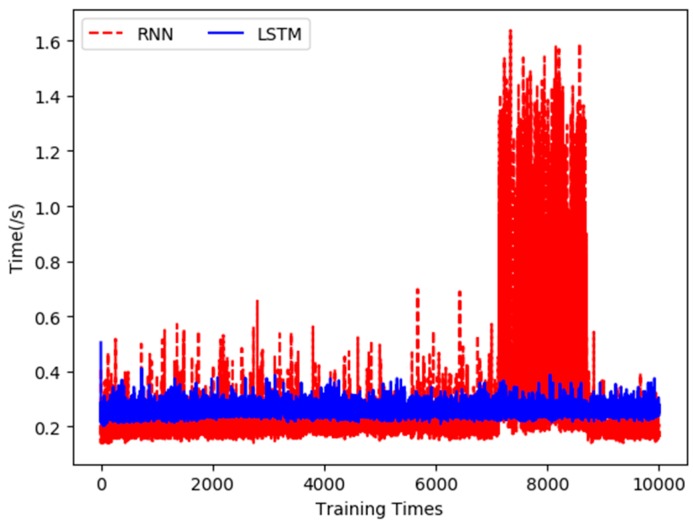
Comparison of training time of water temperature prediction model.

**Figure 14 sensors-19-01420-f014:**
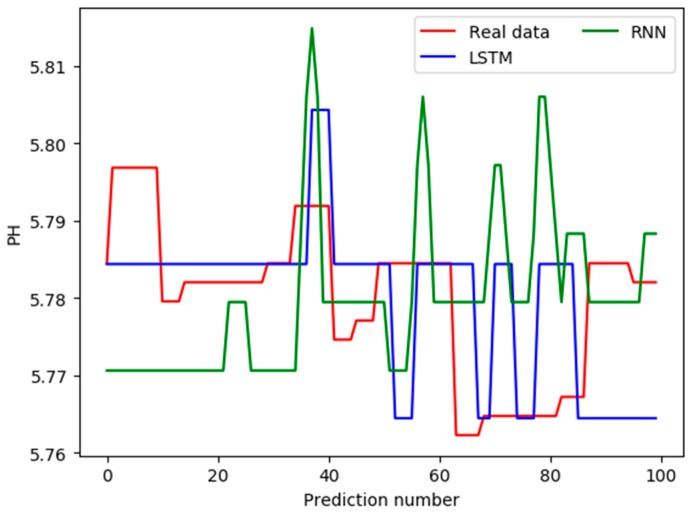
Comparison between predicted values and real values.

**Figure 15 sensors-19-01420-f015:**
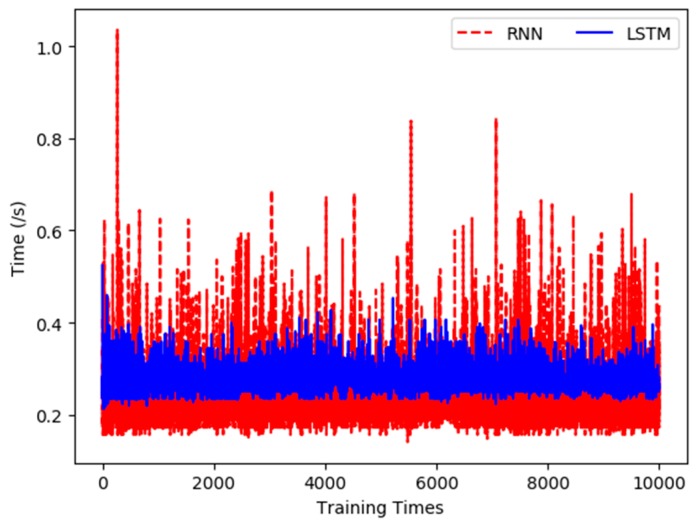
Comparison of training time of pH prediction model.

**Figure 16 sensors-19-01420-f016:**
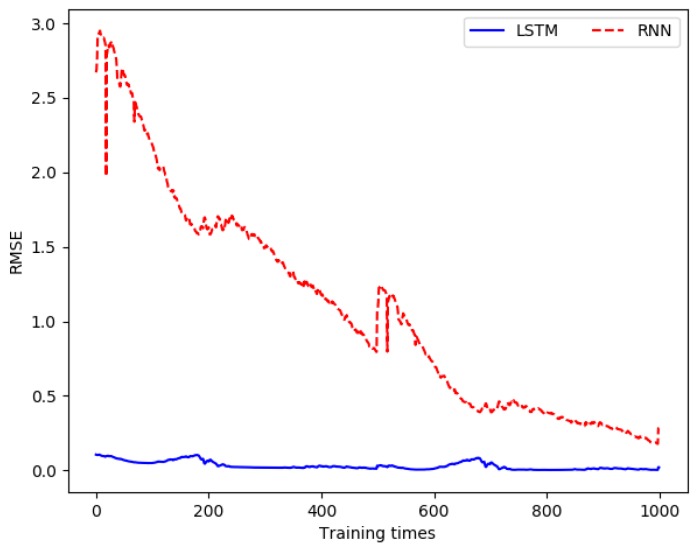
Comparison of RMSE for water temperature.

**Figure 17 sensors-19-01420-f017:**
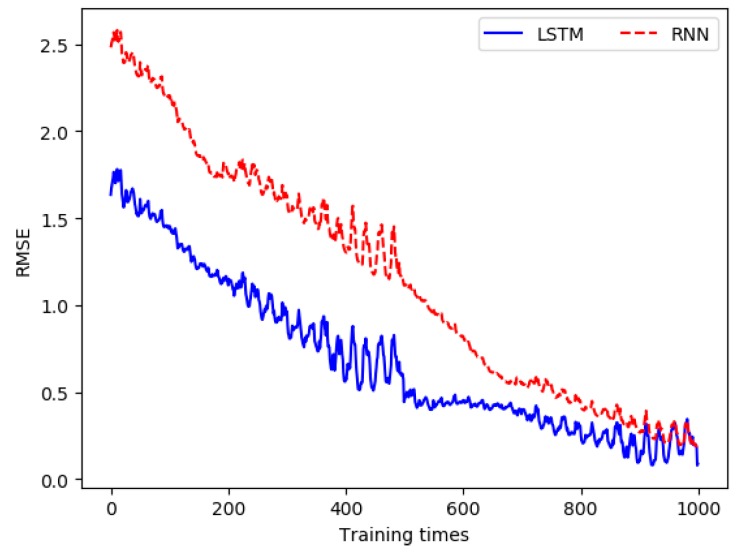
Comparison of RMSE for pH.

**Figure 18 sensors-19-01420-f018:**
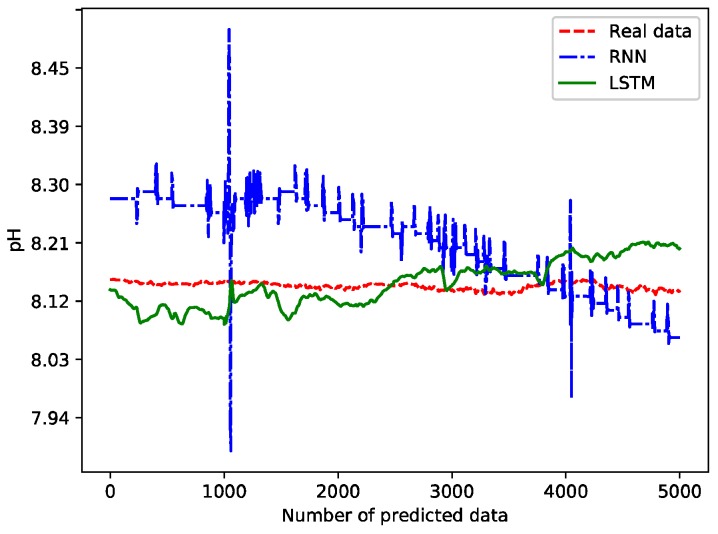
Comparison of the long-term prediction effect for pH.

**Figure 19 sensors-19-01420-f019:**
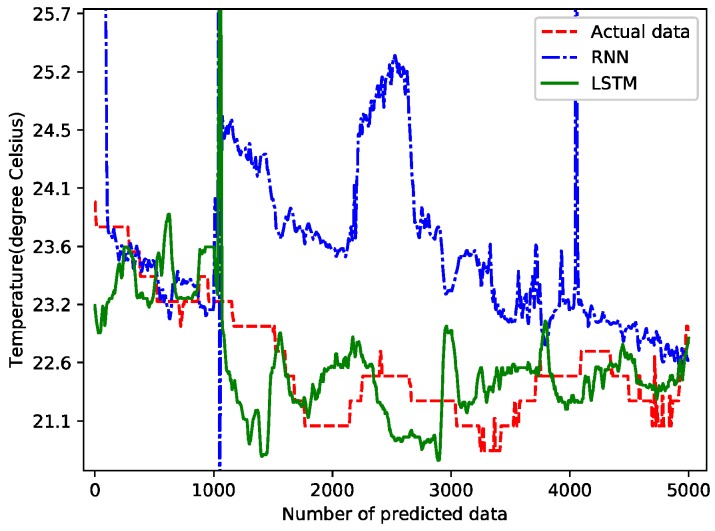
Comparison of the long-term prediction effect for water temperature.

**Table 1 sensors-19-01420-t001:** Calculation results of correlation.

	Conductivity	Salinity	Chlorophyll	Turbidity	Water Temperature	pH	Dissolved Oxygen
pH	−0.9754	0.866538	−0.51448	−0.04556	−0.97497	1	0.197046
Water Temperature	0.999459	−0.91028	0.565859	0.202542	1	−0.97497	0.791544

**Table 2 sensors-19-01420-t002:** Records of MAE, RMSE, and MAPE when training time reached specific value.

Training Time	MAE				RMSE				MAPE			
	Temperature		pH		Temperature(°C)		pH		Temperature		pH	
	LSTM	RNN	LSTM	RNN	LSTM	RNN	LSTM	RNN	LSTM	RNN	LSTM	RNN
1000	0.0149	0.0175	0.0415	0.053	0.0479	0.111	0.113	0.120	0.0455	0.0692	0.0168	0.0239
2000	0.013	0.0156	0.0182	0.0203	0.0222	0.0285	0.0389	0.0457	0.0362	0.0350	0.0032	0.0104
5000	0.0121	0.0143	0.00739	0.00824	0.0168	0.0189	0.0039	0.0051	0.0323	0.0337	0.0013	0.0038
10,000	0.0105	0.0132	0.00274	0.00325	0.0145	0.0155	0.0025	0.0031	0.0289	0.0326	0.0012	0.0023

**Table 3 sensors-19-01420-t003:** The accuracy and range ability of the sensors.

Technical Parameters of Sensors	The Type of Sensors					
	Salinity	Chlorophyll	Turbidity	Water Temperature	pH	Dissolved Oxygen
Range ability	0 ~ 100PSU	0 ~ 400 ug/L	0.1 ~ 1000 NTU	−10 ~ 60°C	0 ~ 14	0 ~ 20.00 mg/L
Accuracy	±1.5% F.S.	±3% F.S.	±1.0 NTU	±0.2°C	±0.01	±1.5%F.S.

**Table 4 sensors-19-01420-t004:** Deviations between the predicted values and the real values using RNN and LSTM.

Group Number	The Deviations between the Real Values and the Predicted Values							
	LSTM	RNN	LSTM	RNN	LSTM	RNN	LSTM	RNN
1	0.132949	3.959442	0.541791	0.414951	2.540453	0.056431	0.198738	0.189815
2	0.400642	3.114111	0.356221	0.503157	2.797719	0.028507	0.147363	0.436588
3	1.036029	3.069812	0.252005	0.396319	3.00332	0.417664	0.333328	0.013683
4	0.961058	2.934413	0.409308	0.658454	2.841458	0.820384	0.482137	0.259189
5	0.728714	2.624992	0.300582	0.833992	2.502212	0.669596	0.216224	1.036307
6	0.519411	2.46428	0.010955	1.187961	1.915524	0.46367	0.146241	0.724474
7	0.459601	2.312542	0.244412	1.439206	1.776256	0.124387	0.371641	0.489075
8	0.768172	2.280364	0.471358	1.513481	1.586839	0.408896	0.699684	0.587774
9	0.78813	2.199178	0.343911	1.485762	1.655596	0.205522	0.63025	0.558739
10	0.505946	2.333197	0.473716	1.770735	1.412733	0.141486	0.368266	0.533711
11	0.246733	2.372123	0.616469	1.727489	0.960963	0.400867	0.447675	0.3642
12	0.059172	2.336197	0.754278	1.390779	0.702773	0.360609	0.683729	0.358777
13	0.232154	2.019102	0.690577	1.144995	0.528385	0.320189	0.98381	0.349546
14	0.010727	2.111242	0.300113	0.267601	0.632107	0.791274	0.86618	0.785739
15	0.280356	2.448614	0.45345	0.289334	0.658988	1.088531	0.561477	1.077822
16	0.303246	2.125633	0.595648	0.666767	0.670855	1.459331	0.49732	1.390808
17	0.289681	2.111117	0.588843	1.139716	0.671857	1.98203	0.668225	2.270634
18	0.236164	2.268004	0.203486	0.983571	0.698048	1.908593	0.831318	2.285945
19	0.276489	2.086995	1.50439	1.087927	0.694925	1.510539	0.684134	2.234941
20	0.394135	2.125502	1.791799	1.889873	0.561602	1.438296	0.460814	2.447277
21	0.412513	2.02414	2.644721	0.006956	0.559784	1.203195	0.446673	2.207668
22	0.344401	1.536572	9.840927	3.26874	0.56474	0.458536	0.649647	1.790473
23	0.467034	1.227761	12.08877	11.93078	0.458013	0.468914	0.662479	1.431288
24	0.567662	0.786095	5.92811	2.601457	0.320005	0.58208	0.633802	0.481562
25	0.541791	0.414951	3.764784	0.738845	0.152873	0.284062	1.006936	0.195481

**Table 5 sensors-19-01420-t005:** Deviations between the predicted values and the real values using RNN and LSTM.

Group Number	The Deviations between the Real Values and the Predicted Values							
	LSTM	RNN	LSTM	RNN	LSTM	RNN	LSTM	RNN
1	0.630802	2.136396	2.855522	0.765329	1.496367	1.962419	2.198166	1.002551
2	0.63822	2.079181	2.614542	0.006417	0.648587	0.551713	2.401637	0.829339
3	0.600423	2.119937	2.505741	0.207085	0.4615	0.475682	2.107028	0.041945
4	0.626563	2.185643	2.340742	0.091206	0.263681	0.433036	1.462349	1.816386
5	1.477082	0.577829	2.309721	0.078737	0.091957	0.539806	0.884081	2.00253
6	1.383567	0.714821	3.950786	1.472333	0.194655	1.105872	0.674252	1.057834
7	1.337259	0.755719	2.171247	0.125621	0.645851	2.272947	1.068124	0.320677
8	1.402000	0.456464	2.064245	0.383645	1.085894	2.573101	1.260291	0.622556
9	1.549016	0.202762	1.961289	0.434519	1.270665	2.070847	1.561966	0.385598
10	1.626345	0.332107	2.175619	1.163942	0.914486	0.769215	0.590714	1.411085
11	0.694443	1.909472	3.731103	3.2055	1.015838	1.065162	0.527791	1.390929
12	0.717155	1.998789	7.108605	11.80848	1.230752	0.973677	0.424462	1.346182
13	0.733518	1.914349	9.976984	13.51213	1.277154	0.954261	0.307454	1.125405
14	0.718865	1.868381	10.86092	8.786944	1.320137	0.931954	0.236954	1.048285
15	0.700548	1.857634	8.690118	2.051635	1.388031	0.921659	0.19318	1.064826
16	0.702013	1.811576	3.038798	1.456259	1.454459	1.149511	0.289509	1.163951
17	0.655123	1.750345	2.099476	0.854766	1.514782	1.228984	0.363189	1.241873
18	0.615315	1.731200	1.680865	0.881667	1.593909	1.266734	0.449198	1.270074
19	0.596754	1.751700	1.390518	0.916591	1.741908	1.105982	0.543807	1.206676
20	1.745037	0.484228	1.026786	0.760142	1.967495	0.593296	0.648737	1.139885
21	1.751331	0.50235	0.747286	1.308068	1.675005	0.524874	0.784917	1.149215
22	1.649274	0.291042	0.516284	1.542153	1.290587	0.158633	0.573094	1.300818
23	1.929368	0.281699	0.112988	1.793935	1.27862	0.197551	0.714104	1.626713
24	2.386151	1.274894	0.426867	1.854815	1.726579	0.699496	0.608600	1.668695
25	2.720645	1.371820	0.886318	1.898942	1.855515	0.974336	0.574483	1.471081

**Table 6 sensors-19-01420-t006:** Records of MAE, RMSE, and MAPE in long-term prediction.

Training Times	MAE				RMSE				MAPE			
	Temperature		pH		Temperature(°C)		pH		Temperature		pH	
	LSTM	RNN	LSTM	RNN	LSTM	RNN	LSTM	RNN	LSTM	RNN	LSTM	RNN
500	0.0421	0.0439	0.0042	0.0325	0.0519	0.5340	0.6236	1.0875	0.085	0.078	0.0092	0.0102
1000	0.0312	0.0424	0.0035	0.0052	0.0457	0.1451	0.3108	0.3254	0.052	0.065	0.0068	0.0073

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
