# Peer review of "A Water Quality Prediction Method Based on the Deep LSTM Network Considering Correlation in Smart Mariculture"

_sensors, 2019, doi:10.3390/s19061420_

Round 1

Reviewer 1 Report

This paper has been fine revised and it is suggested that future work could be addressed in the conclusion.

Author Response

Dear Reviewer,

Thank you very much for your great efforts in reviewing our paper. We reply to your comments below. Please kindly check.

1) This paper has been fine revised and it is suggested that future work could be addressed in the conclusion.

 Response 1:

Thank you very much for your approval of our previous work. The ideas proposed in the conclusion will be tested and verified as soon as possible.

Best Regards

Authors

Reviewer 2 Report

The article “A Water Quality Prediction Method Based on Deep LSTM Network Considering Correlation in Smart Mariculture” by Hu et al. proposes a methodology for predicting pH and water temperature with deep LSTM (Long Short-term Memory) learning networks. The methodology has been tested with data from the Mariculture base at Xincun Town (China) and it has shown lower computational time and better error metrics than the RNN (Recurrent Neural Network) method.  

The objectives of the paper are clear and the methodology follows the state-of-the-art. However, it is required severe improvement on: (i) the execution of such methodology; (ii) the analysis of the results; (iii) the presentation of the paper; (iv) the conclusions of this study.

Hence, despite that the general idea of the article is sounding and it may be within the scope of Sensors, I would recommend rejection. Please find the justification below. Please refer specially to point 5 in General Remarks.

General Remarks

1. It is highly encouraged to revise English by a native speaker. There are sentences hard to read or excessively long (for instance, lines 32-34 or lines 42-43). There are non-English expressions such as “domestic and foreign scholars” (line 66).

2. The Results and Discussion section is somewhat short in comparison with the rest of the paper. Further work would be needed to enhance clarity and justify the proposed method. More discussion about the results and the limits of the method should be addressed. The Conclusions are too generic and refer to points that have not been mentioned throughout the manuscript (see lines 443-445).

3. There are statements without references. Most of the citations are too concise or imprecise. For instance:

Line 32: such general statements need to be reinforced with references.

Line 97: Markov method (12-14). What is “Markov Method”? Hidden Markov Models or Markov Chain Montercalo method?

Line 117: “RNN is usually used to process time sequential data”. Please provide reference.

4. There are paragraphs that would be better in different sections. For instance, lines 198-199 should go to the Introduction. Lines 157 to 160 should go to Results, rather than Methodology.  Lines 321 to 330 may fit better at the beginning of the Methodology section.

5. The execution of the proposed method is my primary concern for suggesting rejection.

There is lacking information about the data gathering. The accuracy of the sensors and the common variation ranges of pH and water temperature at the Case Study are not mentioned in the manuscript.

The training dataset comprises 600 samples * 5 min = 3000 min. Less than 3 days seems a poor representation of the general behaviour of the water temperature or the pH at the Study Area. Is it representative a sample with such low variation of the pH and water temperature at Xincun Town? Is it the fitted LSTM network representative of the different regimes at the Mariculture area? Why? Please justify.

The forecast comprises 100 samples * 5 min = 500 min. So, the forecast is just for the next 12 hours ahead. The variations of temperature (just 0.1ºC) and pH (around 0.02 units) are so low in such a short forecast window, that I find somewhat hard to justify a real world application.

Thus, with such a training dataset and such a poor forecast, it is hardly justifiable the statements made in this contribution. I consider that the Conclusions depend excessively on this limited dataset.

Firstly, I would strongly recommend a longer training dataset (i.e. several months or a whole year as a minimum), in order to enhace robustness. Additionally, I would suggest a longer forecast horizon (i.e. 3 days). Operational process-based forecasts (i.e. based on global and regional ocean circulation models) are providing good predictions for several days ahead (see for instance, https://ocean.weather.gov/SST_fcasts.shtml).  

Specific remarks

1. In order to enhance readability, the flowchart (Figure 8) could go at the beginning of the Methodology section, rather than at the end.  

2. Lines 211-213: Do the authors agree that such conclusions can be found in the state-of-the-art?

3. Lines 217-218: What would be the common tolerance range of pH and water temperature for these mariculture activities?  

4. Line 305: “... the dimension of input data is 4 ...” What kind of variables are used as input?

Figures

I don't understand the aim of the error analysis shown in Figures 6 and 7. With such a small training dataset (600 samples) is not necessary to perform the training 10000 times. Please, indicate the units of RMSE.

Figures 13 and 15: If the point is to remark that LSTM is faster than RNN; then, I would shown the cummulative computational time, rather than the individual time.

Figure 10 and Figure 11 shows that the noise reduction does not affect the training dataset. I don't know what is the accuracy of the sensors; but, depending on the application, the blue time series shown in these Figures can be considered as noisy.

It is hardly encouraged to consider changing the x-axis from Figures 10 to 15. It would enhance clarity to show the time (in seconds, or in hours/days) rather than by the sampling number. That difficults to understand the total time coverage of the training and validation datasets.  

Tables

Table 3 and 4 need to be further explained. What does the 25 group number represents (rows)? What does represent the eight columns? The text hardly mentions such Tables and their implications.

Author Response

Dear Reviewer:

Thank you very much for your great efforts in reviewing our paper. In this revision, we have carefully revised the paper according to the reviewers’ comments. For each comment received, we have prepared a point-to-point response in the attached report. Changes in the revised manuscript are highlighted using yellow. Please kindly check.

Thank you very much again for your editorial efforts in handing/reviewing this paper.

Best Regards

Authors

Reviewer 3 Report

The article presents an interesting application of the TensorFlow library in the field of prediction of water parameters in smart mariculture. The research has been presented in a clear, logical and legible way. The methods based on recurrent neural networks have a number of disadvantages and limitations. Among others, common RNNs cannot be stacked into very deep models. The use of deep LSTM solves this problem. Although the results obtained are still not perfect, the attempt to adapt new methods to known problems deserves appreciation.

After reading the text, I have the following observations:

1. Line 339 - should be Intel(R) instead of Inter(R)

2. Line 340 - do the authors mean PyCharm 2018.3.3? If so, it is not "experimental platform" but Integrated Development Environment (IDE)

3. Did the authors use a higher abstraction layer like Keras for coding?

4. I suggest that the authors consider attaching the source code together with the test data on the github.com website together with references to this webpage in their article. This would greatly enhance scientific value and make it possible for other researchers to test the solution. It would also increase citation.

Author Response

(The authors gave the same response as above.)

Reviewer 4 Report

The manuscript is well written and could have a potential merit. My only suggestion to the authors is to improve the introduction section since there is no need for providing sub-chapters.

Author Response

Dear Reviewer:

Thank you very much for your great efforts in reviewing our paper. We reply to your comments below. Please kindly check.

Comments:

1) The manuscript is well written and could have a potential merit. My only suggestion to the authors is to improve the introduction section since there is no need for providing sub-chapters.

Response 1:

Thank you very much for your approval of our work. We have optimized the structure of the introduction section by removing sub-chapters.

Best Regards

Authors

Round 2

Reviewer 2 Report

The second submission of the article “A Water Quality Prediction Method Based on Deep LSTM Network Considering Correlation is Smart Mariculture” by Hu et al. has successfully addressed most of the suggestions that were made in the previous review. The cover letter has also answered most of my previous concerns.

The overall quality of the paper has significantly increased. I deeply appreciate the effort made by the authors. Hence, my recommendation would be the acceptance of this manuscript, after addressing some moderate issues.

General remarks

My concerns about the proposed methodology have been partly alleviated by the results shown in Section 3.4.

The new Section 3.4 shows that the LSTM method has a better forecasting skill than the RNN (Figures 18 and 19). However, both models (RNN and LSTM) show somewhat noisy forecasts.

The authors have remarked this issue (lines 472-473): "Moreover, compared with the real data, the overall prediction results have strong fluctuation." I strongly agree with this statement. Hence, how would affect these fluctuations to a decision making process? Imagine that some interventions are automated, as for warming up the water or increasing the oxygen concentration under certain thresholds. These spikes could unleash interventions that could harm the mariculture product.

Both LSTM and RNN are handling high frequency data (1 min), so this behaviour can be considered due to the sampling frequency. I would like to find more discussion about this outcome, because it is essential for its applicability. For instance, would it be better to filter these results?

I would recommend to include some post-processing after obtaining the RNN and LSTM time series (blue and green one in Figures 18 and 19). For instance, a low-pass filter, with a cutoff frequency of 5 - 10 mins, may avoid these spikes that do not provide any added value.

2. Another suggestion would be to consider addressing the following content to the coverletter in Section 3.5.:

The authors said: "Different kinds of fish have different tolerance to water quality parameters. Taking saddle-spotted grouper (its scientific name is epinephelus lanceolatus) as an example, the tolerance range of pH value is generally 7.5-9.2, and the range suitable for growth is 7.9-8.4. The water temperature of the growth is 22°C to 30°C, and the minimum tolerance is 15°C and the maximum tolerance is 35°C."

I think that this statement constitutes a relevant part in the leit-motiv of this research. Forecasting is necessary to keep under control certain water quality parameters (i.e. pH and water temperature). With such justification, non-experts in mariculture can easily valuate this contribution.  

3. I would suggest to rewrite the last paragraph (lines 479-483). I would recommend to address both the style and the content. For instance, finishing the manuscript with "... method and so on" (line 483) seems somewhat innapropiate. Some references about applications using PCA and SVD would be highly recommended. It is hard to evaluate, from a reader point of view, why the authors suggested PCA and SVD just at the end of manuscript. How would the proposed methodology combine these additional techniques?

4. The conclusions (lines 493-496) highlight some statements that have not been described elsewhere in the paper. The Conclusions summarise the outcomes of the paper, but it cannot add new information. I would suggest to firstly introduce these results at the Discussion section.

Specific remarks

Line 419: Please consider revising "the future tend" by "the future trend".

Author Response

Dear Reviewer:

Thank you very much for your great efforts in reviewing our paper. In this revision, we have carefully revised the paper according to the reviewer’ comments. For each comment received, we have prepared a point-to-point response in the attached report. Changes in the revised manuscript are highlighted using light gray. Please kindly check.

Thank you very much again for your editorial efforts in handing/reviewing this paper.

Best Regards

Authors

Round 3

Reviewer 2 Report

The third submission of the article "A Water Quality Prediction Method Based on Deep LSTM Network Considering Correlation in Smart Mariculture" by Hu et al. has successfully addressed all the suggestions that were made in the second review. The cover letter has also answered my previous concerns.

The overall quality of the paper has increased. I have enjoyed this reading and I deeply appreciate the effort made by the authors. It is consistent and realistic with the current drawbacks of the LTSM. Hence, my recommendation would be the acceptance of this manuscript.

I would like to wish success to the authors with their forthcoming research.

My most sincere congratulations!

This manuscript is a resubmission of an earlier submission. The following is a list of the peer review reports and author responses from that submission.

Round 1

Reviewer 1 Report

Dear Authors,

With great regret I inform you that I reject the manuscript . There is almost no any connection between smart mariculture efficiency and the results you present, or you did not explained well. Many trivial formulas, unnecessary tables, trivial conclusions, no necessary explanations etc. make the manuscript unclear and chaotic. 

Best regards,

Reviewer

Author Response

Dear Reviewer:

Thank you very much for your great efforts in reviewing our paper. 

In this revision, we have substantially revised the paper. Changes in the revised manuscript are highlighted. Please kindly check.

Thank you very much for your editorial efforts in handing/reviewing this paper.

Best Regards.

Authors

Reviewer 2 Report

The authors propose a novel water quality prediction method based on correlation Long Short-term Memory deep learning. The literature survey is sufficient, but the methodology is weak. The authors do not discuss the major limitations of the model and fail to carry out an extensive discussion. My other comments are as follows:

Main comments:

1.     Why were other interpolation techniques deemed far efficient than linear interpolation was not tested? How much can spline/nearest-neighbor/cubic interpolations improve the missing dataset?

2.     Figure 9/10: It is better to plot just the difference (bias). Or use scatter plot. It is hard to see the difference in Figure 9/10.

3.     Table 3 and 4 are unclear? Each Row and Column need to be properly labeled?

4.     Why not use other metrics (RMSE/MBE/MAE etc.) to evaluate the predictions?

5.     Why is RNN the benchmark? Are there any simplistic statistical models that can be tested as well?

6.     There is no sensitivity analysis conducted to the number of inputs used or errors within measurements?

7.     Is using 610 and 100 groups of dataset sufficient for such analysis?

8.     How do you test the robustness of the new model? Can it capture extreme values?

9.     How do you avoid overfitting?

Author Response

Dear Reviewer:

Thank you very much for your great efforts in reviewing our paper. The detailed comments have helped us a lot in improving both the quality and the clarity of the paper.

In this revision, we have substantially revised the paper according to the reviewers’ comments. For each comment received, we have prepared a point-to-point response in the attached report. Changes in the revised manuscript are highlighted. Please kindly check.

Thank you very much for your editorial efforts in handing/reviewing this paper.

Best Regards

Authors

Reviewer 3 Report

The authors present an interesting approach for water quality prediction based on LSTM and deep neural networks. This is of high interest for the general reader and it is shown its applicability. There is a number of improvements the authors should address. As some of them are of relative importance I'm proposing major revision.

At the abstract, the authors talk about deepLSTM with no explanation further about what they refer when introduce the word "deep" in the text. Also at the abstract, the authors write about "experimental results" with no mention further what are those experiments.

At the introduction, 

- the authors mention some items of the bibliography in the text using the format as the following: "Jin, C. et al."; I would suggest to just write the surname "Jin et al." and do it at all cases the authors wish to specifically mention the cited authors.

- the authors should make a double effort on the proofreading of the text: Line 62 "It combines..." Line 63 "Lots of researchers..." Line 85 "large time..." and so forth over this section and the entire manuscript.

- the authors refer to "the simple mathematical model" but don't explain what exactly it is (line 68)

- the authors should include additional citations plus, perhaps, brief explanation when introducing technical jargon such as deep neural networks (line 90) or RNN (line 103).

- importantly, the authors should state clearly the advantages/disadvantages of adopting their methodology (also in Conclusions supported by some results).

At Section2,

- please, explain better what it means "change slowly and excessively" (line 137)

- Please use present tense at line 139

- The authors should specify the criteria of Table 1 (why and/or add citation)

- The authors should precise the information of packages, libraries and computer configuration at line 245 (or 252)

- Figures 6 and 7, add labels to both X-axis and units to Y-axis labels ... and so on... please, double check labels of all the figures in the manuscript

- importantly, the authors should better detail the experimental study. Data available, frequency in which data is collected etc.

- also, the authors should provide a better "picture" of the whole process, perhaps by adding a flowchart describing the overall process.

At Section3,

- Unify the way to write numbers: 10000 and 10,000 in the same paragraph (lines 333 and 335)

Section Conclusions

- It is quite short section. The authors should develop more the conclusions, promote discussion and propose future research

Author Response

Dear Reviewer:

Thank you very much for your great efforts in reviewing our paper. The detailed comments have helped us a lot in improving both the quality and the clarity of the paper.

In this revision, we have substantially revised the paper according to the reviewers’ comments. For each comment received, we have prepared a point-to-point response in the attached report. Changes in the revised manuscript are highlighted. Please kindly check.

Thank you very much for your editorial efforts in handing/reviewing this paper.

 Best Regards.

Authors

Round 2

Reviewer 1 Report

Dear Authors,

The text from line 184 (from It is shown…) to line 193 as well as Table 1 and Formula 4 should be necessarily removed. This is trivial information good for schoolbooks not for scientific paper. The more important is to stress that Pearson’s correlation coefficient has many important shortcomings, especially when environment is studied. In general standards correlation or regression methods are nowadays often replaced by more advanced integration method as spatial cross-correlations or cokriging. It seems that you do not know the recent advances in this subject.

You added some references making first revision, well, however they not describe the disadvantages of Pearson’s correlation coefficient in environmental studies, e.g.  of lakes, and they do not take into account spatial context.  To minimize this shortcoming at least, please read carefully and consider to refer in the introduction to the work: “Geostatistical study of spatial correlations of lead and zinc concentration in urban reservoir. Study case Czerniakowskie Lake, Warsaw, Poland, P. Fabijańczyk, J. Zawadzki, M. Wojtkowska, Open Geosciences, 2016, 8 (1), 484-492” where spatial cross-correlations (cross-semi-variances) were used to describe water quality parameters in small lake, and overcome at the same times the week sides of Pearson’s correlation coefficient  This will be surely profitable for the potential readers interested in such problems.

Best regards.

Reviewer

Author Response

Dear Reviewer:

Thank you very much for your great efforts in reviewing our paper. In this revision, we have carefully revised the paper according to the reviewers’ comments. For each comment received, we have prepared a point-to-point response in the attached report. Changes in the revised manuscript are clearly highlighted using the "Track Changes" and annotation functions of Microsoft Word. Please kindly check.

Thank you very much again for your editorial efforts in handing/reviewing this paper.

Best Regards.

Authors

Reviewer 3 Report

The authors suitable address a good number of the issues I raised in my previous report. However, I'm sorry to say that the figures should be modified to add the units of the measures in the axis labels and do not explain them in the text. Not sure if this is minor or major revision again but from my point of view it is important to improve the paper quality.

Author Response

Dear Reviewer:

Thank you very much for your great efforts in reviewing our paper. In this revision, we have carefully revised the paper according to the reviewers’ comments. For each comment received, we have prepared a point-to-point response in the attached report. Changes in the revised manuscript are clearly highlighted using the "Track Changes" and annotation functions of Microsoft Word. Please kindly check.

Thank you very much again for your editorial efforts in handing/reviewing this paper.

Best Regards

Authors

Round 3

Reviewer 3 Report

The authors missed to write the units to some of the axis in the figures. The figures labels also seem to have different font sizes. A better English proofreading is required to replace some few awkward expressions.

Author Response

Dear Editor:

Thank you very much for your great efforts in reviewing our paper. 

In this revision, we have carefully revised the paper according to the reviewers’ comments. For each comment received, we have prepared a point-to-point response in the attached report. All changes in the revised manuscript are clearly highlighted using the "Track Changes" function of Microsoft Word and dark blue color. Please kindly check.

Thank you very much again for your editorial efforts in handing/reviewing this paper.

Best Regards.

Authors
